# Antenatal care component utilization and associated factors among pregnant women in Ethiopia: Multilevel analysis of Ethiopian Mini Demographic and Health survey 2019

Desalegn Shiferaw[1,2]*, Bikila Regassa Feyisa[1,3], Bayise Biru[3,4], Mubarek Yesse[1,5]

1 Department of Epidemiology, Faculty of Public Health, Jimma University, Jimma, Ethiopia, 2 Department of Public Health, Institute of Health Sciences, Dambi Dollo University, Dembi Dollo, Ethiopia, 3 Department of Public Health, Institute of Health, Wallaga University, Nekemte, Ethiopia, 4 Department of Human Nutrition and Dietetics, Faculty of Public Health, Jimma University, Jimma, Ethiopia, 5 Department of Public Health, College of Medical and Health Science, Werabe University, Werabe, Ethiopia

* latisenako@gmail.com

**Data Availability Statement:** The data we used in this study is available in public repository https://dhsprogram.com

## Abstract

### Introduction

Maternal and neonatal health are among the top prioritised agendas of global health care with due emphasis given to developing countries, where the burden is profound. Antenatal care accompanied by its recommended components is highly beneficial for both maternal health and birth outcome.

### Objective

The objective of this study was to identify the proportion of pregnant women who received adequate Antenatal care components and associated factors among Ethiopian women.

### Methods and materials

We used a nation-wide data from Mini Ethiopian Demographic and Health Survey (MEDHS) of 2019. All women of age 15–49 and who had at least one ANC visit, who were either permanent residents of the selected households or visitors who slept in the household the night before the survey, were eligible to be interviewed. Since we utilised multilevel logistic regression model, the STATA output had two components, the fixed effect and the random effect. In our model, the fixed effect part was displayed by odds ratio while the random effect was addressed by variance and intra-cluster correlation (ICC).

### Results

From the total women with at least one antenatal care (ANC) visit 55.41% (95% CI 53.60%, 57.20%) of them received adequate components of the care. In the final model after adjusting for the cluster and individual level variables, attending primary (AOR = 1.45; 95% CI: 1.15 to 1.84), secondary (AOR = 2.21; 95% CI: 1.51 to 3.24) and higher education (AOR =

**Funding:** The authors received no specific funding for this work.

**Competing interests:** The authors have declared that no competing interests exist.

2.42; 95% CI: 1.38 to 4.26) were significantly associated with higher odds of receiving adequate components of ANC. Similarly, wealth index of middle (AOR = 1.51; 95% CI: 1.06, 2.14), richer (AOR = 1.92; 95% CI: 1.32, 2.80), and richest (AOR = 3.86; 95% CI: 2.35, 6.33) compared to poorest index and having two or more ANC visits were significantly associated with receiving adequate components of ANC. On the other hand, being from Oromia region, from female headed household and protestant religion were negatively associated with receiving adequate components of ANC.

## Conclusion

The proportion of women who received adequate ANC component was much lower compared to the universal recommendation for every woman on ANC visit. Educational status, wealth index, number of ANC visit, region of residence and type of health facility were significantly associated with the odds of receiving adequate components of ANC. The government should pay attention to those without any formal education, encouraging pregnant women to receive the optimum number of ANC visits, and devising techniques to address those in poorest wealth index so that the proportion of adequate components of ANC will be increased.

## Introduction

Maternal and neonatal health are among the top prioritized agendas of health care globally with due emphasis given to developing countries, where the burden is profound [1, 2]. Ethiopia is among the countries with slow decline in maternal death and it is a recent memory that the country failed to meet the Millennium development target of reducing maternal mortality [3]. The country, supported by different developmental partners, is tremendously doing in improving maternal health services to reduce maternal mortality [3]. As one of the sustainable development goal targets, reducing the maternal mortality ratio has been getting special attention from the Ethiopian government and different stakeholders to achieve the target [3].

Irrespective of the on-going aggressive works to reduce maternal mortality toward the target figure of sustainable development goal (SDG, 70/100,000 live births, it will be challenging for most African countries to meet the target [1, 4, 5].

Existing evidence shows that Antenatal Care (ANC) is an appropriate gateway for the full package of maternal health care services [6] and those with good adherence to ANC visits are more likely to stick to the remaining services in the continuum of maternal health care services compared to those with poor compliance [7, 8]. On the other hand, ANC visits should be accompanied by provision and utilization of the essential components which are said to be the building blocks of the care in ensuring its quality [9]. According to a study conducted in Northwest Ethiopia, less than three in a hundred pregnant women received adequate ANC components in the study [10]. Many of the studies conducted in Ethiopia focused on a mere number of ANC visits while the components of the services expected to be delivered during each visit are overlooked [11, 12]. There are several components of ANC provided during each visit and used to measure the quality of ANC [13–17]. To make the efforts to improve ANC services impactful, due emphasis should be given to the components of care during each visit [18]. These components include blood pressure measurement, taking blood and urine samples, nutritional counselling, provision of iron/folic acid, and others [6, 15, 16]. Even though

different researchers consider a variable number of components, most of the demographic and health surveys including the 2019 mini Ethiopian Demographic and Health Survey considered four of the components [15].

Any antenatal care visit in the absence of adequate components may not bring the intended benefits [6]. Although few pocket studies were done to address this particular topic, most of them lack representativeness. On the other hand, representative data like that of EDHS give reliable and timely evidence regarding the problem to convince policymakers who finally translate it into action. Therefore, the current study aimed to identify the proportion of women who received adequate components of ANC using the country-level data from the mini Ethiopian Demographic and Health Survey of 2019.

## Methods and materials

### Source of data

The study used nationwide data from Ethiopian the Mini Demographic and Health Survey (EMDHS) 2019 [19]. The survey was conducted by the Ethiopian Public Health Institute in collaboration with the Ethiopian Ministry of Health (MoH), Central Statistical Agency (CSA), and United States Agency for International Development (USAID) from 21st March 2019 - 28th June 2019.

The 2019 mini-EDHS employed a stratifying sampling technique with two stages and women who were permanent residents of the selected household and visitors who spent a night before the interview with the family were interviewed.

An enumeration area (EA) is a geographic area that addresses an average of 131 households. The sampling frame contains information about EA location, type of residence (urban or rural), and estimated number of residential households.

In the first stage, 305 enumeration areas including 93 from urban and 212 from rural were selected with probability proportional to EA size and with independent selection in each sampling stratum. During the second stage, a fixed number of 30 households per cluster were selected with an equal probability of systematic selection.

The 2019 MEDHS data were pre-tested before the actual data collection. Data collectors had received training in interviewing techniques, field procedures, the content of the questionnaires, and how to administer both paper and electronic questionnaires; after all, questionnaires were finalized in English, and then translated into Amarigna, Tigrigna, and Afan Oromo. The sample considered for this analysis consisted of women aged 15–49 years who had a history of pregnancy in the last five years and had at least one ANC visit which was 2918 women.

### Study population

All women aged 15–49 and who have at least one ANC visit and either permanent residents of the selected households or visitors who slept in the household the night before the survey, were eligible to be interviewed. In all selected households, women aged 15–49 years were interviewed using the Woman's Questionnaire.

### Study variables

**Outcome variable.** The outcome variable in this study was a receipt of the four components of ANC among women with at least one ANC visit during the last pregnancy. This outcome variable was dichotomized into "Yes" if the woman received the four components and "No" if she missed at least one of the components. These components include measuring blood pressure, taking blood samples, taking urine samples, and whether she was given/bought

iron tablets/syrup. The above listed components were considered irrespective of order of the antenatal care visits (they can be at the first, second, or at any time of the visits).

**Independent variables.** The independent variables were women's age, educational level, timing of the first ANC visits, number of ANC visits, birth order, wealth index, current marital status, place of residence, region, and type of health facility. There are two levels of independent variables namely community level and individual level variables. The community-level variables are Residence (urban and rural), Region of residence (Tigray, Afar, Amhara, Oromia, Somali, Beneshangul, SNNPR, Gambela, Harari, Addis Ababa, and Dire Dawa), and type of health facility in their locality. The remaining variables are individual level, and they include the age of the mother which is grouped as 15–19, 20–24, 25–29, 30–34, 35–39, 40–44, and 45–49 years. Educational status is categorized as no formal education, primary (1–8), secondary (9–12), and higher (college and higher education). The wealth index was categorised as poorest, poorer, middle, richer, and richest. Religion of the participants was Orthodox, Catholic, Protestant, Muslim, traditional, and others. The other independent variable was current marital status which was categorised as never in union, married, living with a partner, widowed, divorced and separated. The number of ANC was grouped as one, two, three, and four or more times. The sampling procedure of the eligible women is shown in figure one (Fig 1).

## Data processing

STATA version 14.0 was used for data analysis. Before proceeding with further analysis, we weighed all the distributions/variables to keep the representativeness of the demographic and health survey data and for this; we used (v005/1000000).

As portrayed in the sampling procedure above (Fig 1), we identified 2918 eligible women (unweighed) was and 2913 after applying weighting. These women were those with at least one ANC visit from the total 3979 women who were pregnant/gave birth five years preceding the survey.

Since we utilized a multilevel logistic regression model, the STATA output had two components, the fixed effect and the random effect. In our model, the fixed effect part was displayed by odds ratio while the random effect was addressed by variance and intra-cluster correlation (ICC). We fitted different models including the null model (with no independent variables), which was used to see a difference in receiving ANC components at the cluster level. The other models were model one where we checked for individual level independent variables, model two in which we included cluster level or community level variables and model three in which we adjusted for both the individual and community level independent variables.

## Statistical analyses

Our dependent variable is whether the pregnant woman received all the components of ANC considered in the study (measuring blood pressure, taking blood samples, taking urine samples, and whether she was given/bought iron tablets/syrup), which was categorized as adequate or not adequate and multilevel logistic regression model was applied for further analysis. For the sake of simplicity, those who received adequate components were labelled as "Yes" and those who received in adequate were labelled as "No". Multilevel logistic regression is a statistical method used to analyse data that has a hierarchical or nested structure, such as when individuals are nested within groups or organizations. It is an extension of logistic regression, which is used to model binary outcomes, such as whether or not pregnant women have adequate ANC components. In multilevel logistic regression, the model allows for the estimation of both the within-group variation (level 1) and between-group variation (level 2). This approach can improve the accuracy of estimates and account for the dependencies or

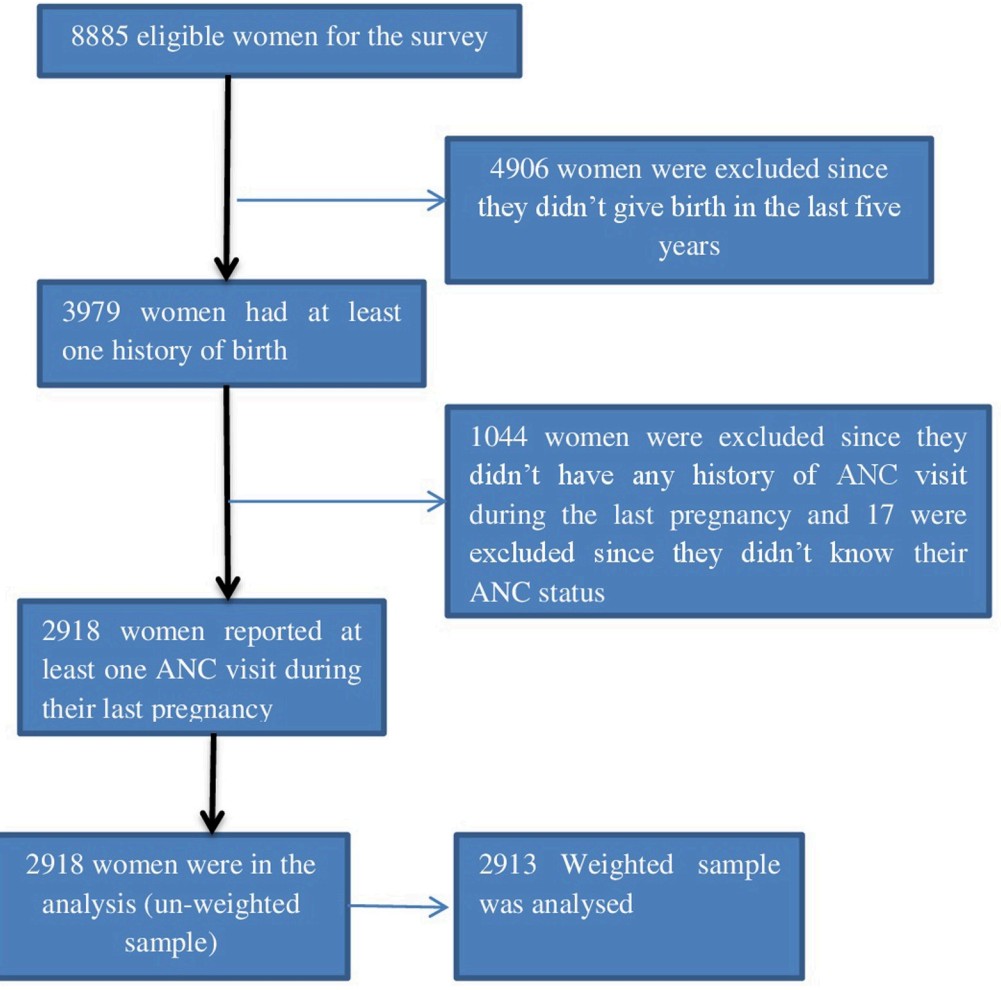

**Fig 1. Sampling procedure of the eligible women.**

correlations that may exist within the data. It provides insights into how individual and group-level factors influence the outcome of interest.. In this study, women are nested within the 303 clusters and there is dependence across levels of nested data since they share common characteristics. To avoid such problems, a multilevel logistic regression model was selected as the appropriate model. Such a regression model is used to analyse sources of variability in multilevel data by considering variability at each level. The null model involves no independent variable specified at either level. The next model is a model that includes individual-level predictors and the third model is the one fitted with community (cluster) level variables. The final model involves random intercept and explanatory variables from the two levels.

## Model selection

Based on the hierarchical nature of the data and considering that the dependent variable is binary, we selected multilevel logistic regression model. During fitting for multilevel logistic regression model, we started with the null model in which there was no any independent variables to check if there is cluster effects in their variability. Accordingly, the null model indicated that there is a significant difference in odds of receiving ANC components at community level which was confirmed by variance of 1.28 and intra-cluster correlation (ICC)

of 27.99% which was attributed to community level difference from the total variation. Then we fitted models for individual level variables (Model II), community level model (Model III) and the final model (Model IV) that adjusted for both the community level and individual variables. From the fitted models the final model was with the highest log likelihood -1578.54. Based on this, the final model (Model IV) was selected as best fitted model to see the association between the dependent variable and the independent variables (Table 1).

The final multilevel model can be written as:

$$L_{mlm}(X, Y) = -\sum_{n=1}^{N} W_n \left( \sum_{i=1}^{|V|} Y_i^n ln(f_s(X)_i^n) \right)$$

## Ethical considerations

The permission for the current study was obtained through a written letter from the DHS program after the submission of a viable research title and a summary of the statement of the problem. The dataset we used is available in a public repository (https://dhsprogram.com) and can be accessed up on request. The data we used are freely available, without any personal identifiers and the data can't be related to individual participants. Beyond this, all the ethical concerns regarding the Ethiopia Mini Demographic and Health survey were mentioned in the report [19].

## Results

A total of 2913 women who gave birth five years before the survey and had at least one antenatal care visit were included in this study. From the mothers included for the final analysis in this study, 945 (32.45%) were in the age range of 24–29 years and 2,048 (70.31%) of them were rural resident. About 94% of the women were married and 1800 (61.79%) have reported that their first visit for ANC was after three months of their pregnancy (Table 2).

### Proportion of women with adequate ANC components

Among those women with at least one ANC visit, 55.41% (95% CI 53.60%, 57.20%) received an adequate ANC components. Of these respondents, 88.10% received blood pressure measurements while 73.85% got their urine sample taken. Additionally, 78.83% of them got their blood sample taken, and 77.18% of the women were given/prescribed iron tablets/syrup during their visit. Surprisingly, approximately 4% of women did not receive any of these essential ANC components during their visits (Fig 2).

**Table 1. Model diagnosis for antenatal care component utilization: Multilevel logistic regression, 2023.**

| Parameters | model I | model II | model III | Model IV |
|---|---|---|---|---|
| ICC (%) | 27.99 | 24.23 | 20.64 | 20.81 |
| Log likelihood | -1750.22 | -1618.70 | -1681.71 | -1578.54 |
| AIC | 3504.43 | 3303.40 | 3399.43 | 3255.08 |
| BIC | 3516.39 | 3500.70 | 3507.04 | 3548.03 |
| Deviance(-2log likelihood) | 3500.44 | 3237.40 | 3363.42 | 3157.08 |

**Table 2. Characteristics of women who gave birth in the last five years in Ethiopia, 2023.**

| Variable (n = 2918) | Category | Frequency (%) Un-weighted | Frequency (%) Weighted |
|---|---|---|---|
| Maternal age | 15–19 | 169 (5.79) | 153 (5.25) |
| | 20–24 | 611 (20.94) | 604 (20.73) |
| | 25–29 | 948 (32.49) | 945 (32.45) |
| | 30–34 | 600 (20.56) | 599 (20.58) |
| | 35–39 | 392 (13.43) | 408 (14.02) |
| | 40–44 | 153 (5.24) | 156 (5.35) |
| | 45–49 | 45 (1.54 | 48 (1.64) |
| Residence | Urban | 889 (30.47) | 865 (29.69) |
| | Rural | 2,029 (69.53) | 2,048 (70.31) |
| Region | Tigray | 322 (11.03) | 270 (9.27) |
| | Afar | 230 (7.88) | 32 (1.10) |
| | Amhara | 336 (11.51) | 711 (24.41) |
| | Oromia | 346 (11.86) | 1,076 (36.94) |
| | Somali | 86 (2.95) | 63 (2.16) |
| | Beneshangul | 299 (10.25) | 39 (1.34) |
| | SNNPR | 326 (11.17) | 558 (19.16) |
| | Gambela | 268 (9.18) | 16 (0.55) |
| | Harari | 247 (8.46) | 9 (0.31) |
| | Addis ababa | 227 (7.78) | 121 (4.15) |
| | Dire dawa | 231 (7.92) | 18 (0.62) |
| Educational status | No education | 1,269 (43.49) | 1,278 (43.88) |
| | Primary | 1,066 (36.53) | 1,150 (39.46) |
| | Secondary | 356 (12.20) | 333 (11.43) |
| | Higher | 227 (7.78) | 152 (5.23) |
| Wealth index | Poorest | 594 (20.36) | 397 (13.64) |
| | Poorer | 503 (17.24) | 587 (20.15) |
| | Middle | 457 (15.66) | 588 (20.20) |
| | Richer | 460(15.76) | 576(19.77) |
| | Richest | 904 (30.98) | 765(26.26) |
| Religion | Orthodox | 1,089 (37.32) | 1,217 (41.77) |
| | Catholic | 19 (0.65) | 6 (0.21) |
| | Protestant | 585 (20.05) | 793 (27.21) |
| | Muslim | 1,202 (41.19) | 872 (29.94) |
| | Traditional | 15 (0.51) | 21 (0.72) |
| | Other | 8 (0.27) | 4 (0.14) |
| Current marital status | never in union | 19 (0.65) | 13 (0.45) |
| | married | 2,675 (91.67) | 2,726 (93.59) |
| | living with partner | 25 (0.86) | 19 (0.65) |
| | widowed | 32 (1.10) | 26 (0.90) |
| | divorced | 113 (3.87) | 91 (3.11) |
| | separated | 54 (1.85 | 38 (1.31) |
| Birth order | 1 | 705(24.16) | 688 (23.62) |
| | 2–4 | 1372 (47.02) | 1365 (46.86) |
| | ≥5 | 841 (28.82) | 860 (29.52) |

(*Continued*)

**Table 2.** (Continued)

| Variable (n = 2918) | Category | Frequency (%) Un-weighted | Frequency (%) Weighted |
|---|---|---|---|
| Number of ANC | 1 | 141 (4.83) | 130 (4.47) |
| | 2 | 353 (12.10) | 294 (10.09) |
| | 3 | 768 (26.32) | 801 (27.50) |
| | ≥4 | 1,656 (56.75) | 1688 (57.96) |
| Timing of ANC visit in months | ≤3 | 1245 (42.67) | 1088 (37.35) |
| | >3 | 1649 (56.51) | 1800 (61.79) |
| | Not known | 24 (0.82) | 25 (0.86) |

### Factors associated with receiving ANC contents

In the final model, after adjusting for community and individual level variables, educational status, wealth index, number of ANC, region of residence, and type of health facility were found to be significantly associated with receiving the components of ANC. Specifically, the odd of receiving adequate ANC components was lower by 58% (AOR = 0.42; 95% CI: 0.21 to 0.81) among women in Oromia compared to women from the Tigray region.

The level of education was found to be significantly associated with receiving adequate components of antenatal care (ANC). Women with primary education had higher odds of receiving adequate ANC components compared to those without any formal education

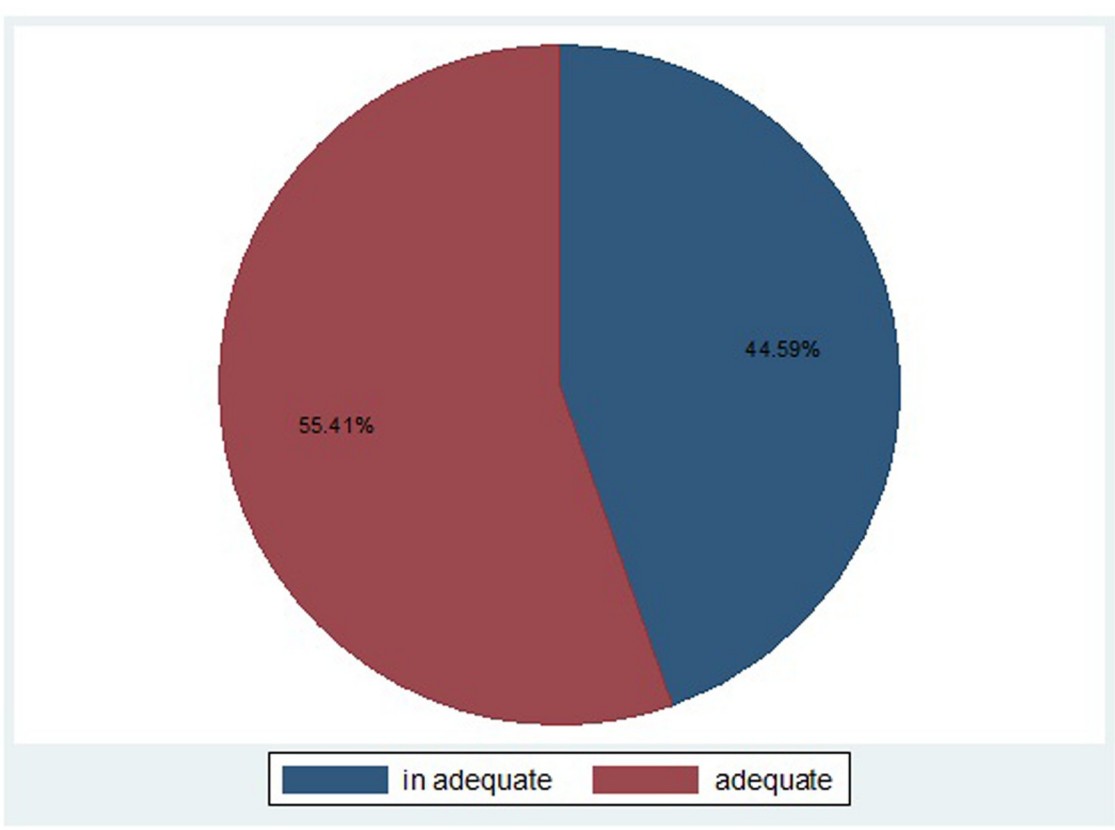

**Fig 2. Proportion of pregnant women received adequate ANC components.**

(AOR = 1.45; 95% CI: 1.15 to 1.84). Similarly, women with secondary education (AOR = 2.21; 95% CI: 1.51 to 3.24) and higher education (AOR = 2.42; 95% CI: 1.38 to 4.26) had increased chances of receiving adequate ANC components compared to women without any formal education. Furthermore, wealth index was the other variable found significantly associated with the outcome variable. Women from the middle (AOR = 1.51; 95% CI: 1.06to 2.14), richer (AOR = 1.92; 95% CI: 1.32to 2.80), and richest (AOR = 3.86; 95% CI: 2.35to 6.33) wealth index categories had higher odds of receiving adequate components of ANC compared to those from poorest wealth index category. Number of ANC visits was the other variable showed significant association with receiving adequate components of ANC in this study. Accordingly, women with two ANC visits (AOR = 3.11; 95% CI: 1.70 to 5.68), three ANC visits (AOR = 5.46; 95% CI: 3.12to 9.55, and four or more ANC visits (AOR = 6.44; 95% CI: 3.70to 11.21) were more likely to receive adequate components of ANC compared to those with only one antenatal care visit.

The type of health facility has shown statistically significant association with receiving adequate components of ANC. Those who received ANC from government hospital (AOR = 3.07; 95% CI: 1.86to 5.08), from governmental health centre (AOR = 2.52; 95%CI: 1.72to 3.68), and non-governmental health facility (AOR = 4.23; 95% CI: 1.65to 10.85) had higher odds of receiving adequate ANC components compared to their counterparts.

Sex of head of the household was the other independent variable which showed a significant association with the outcome variable. The probability of receiving adequate ANC components was lower among women from households headed with female (AOR = 0.54; 95% CI: 0.39 to 0.76) compared with male headed households. On the other hand, Protestant women were less likely to receive adequate components of ANC compared to Orthodox Christian women (AOR = 0.62; 95% CI: 0.41 to 0.92) (Table 3).

## Discussion

This study aimed to assess the extent of antenatal care components utilised and its associated factors among women who had at least one ANC visit during their recent pregnancy. In this study, we considered only four of the intended components of ANC (blood pressure measured, blood sample taken, urine sample taken and iron tablet/syrup prescribed). According to World Health Organization, every pregnant woman on ANC visit is expected to receive the recommended components of the service. This study showed that more than half of the women had received adequate ANC components and region of their residence, educational status, wealth index, sex of household head, number of ANC visit and type of health facility were significantly associated with receiving ANC components.

The level of ANC components a pregnant mother receives during a routine ANC visits serves as the quality indicator for the intended health care service the mother deserves during the her pregnancy period. Among others, blood pressure measurement (88.10%) and blood sample test (78.83%) were frequently utilised. On the other hand, iron/folic acid tablet (77.18%) and urine sample test (73.85%) were reported by respondents. These components are intended for all the pregnant mothers received ANC service. On aggregation, the weighted proportion of women who received adequate ANC components to be 55.41%. This figure is higher than the finding of study conducted in ten low and middle income countries that ranges from 10% in Jordan to 50% in Nepal, Nigeria, Colombia and Haiti [20], studies conducted in east African countries (11.16%) [17], Ethiopia (22.48%) [15], Bangladesh (22%) [14], Zambia (47.1%) [21], Nepal (43%) [22], Uganda (18.4%) [23]. Furthermore, it is higher than the findings of some studies conducted in [15, 16, 24]. This difference could be because our current study measured the adequacy of the service by considering only four components while other studies included more

**Table 3. Factors associated with received ANC contents among child-bearing age women in Ethiopia, 2023.**

| Variable (n = 2918) | Category | Null Model | Model II | Model III | Model IV |
|---|---|---|---|---|---|
| | | | AOR (95% CI) | AOR (95% CI) | AOR (95% CI) |
| Maternal age | 15–19 | | 1 | | 1 |
| | 20–24 | | 1.01(0.65, 1.59) | | 0.93(0.59, 1.46) |
| | 25–29 | | 0.95(0.59, 1.53) | | 0.88(0.55, 1.43) |
| | 30–34 | | 0.85(0.51, 1.43) | | 0.78(0.46, 1.32) |
| | 35–39 | | 0.77(0.44, 1.36) | | 0.73(0.41, 1.29) |
| | 40–44 | | 0.96(0.51, 1.81) | | 0.88(0.46, 1.67) |
| | 45–49 | | 0.64(0.27, 1.48) | | 0.61(0.26, 1.44) |
| Residence | Urban | | | 1 | 1 |
| | Rural | | | 0.54(0.34, 0.84) | 1.27(0.76, 2.13) |
| Region | Tigray | | | 1 | 1 |
| | Afar | | | 0.66(0.24, 1.81) | 1.05(0.34, 3.28) |
| | Amhara | | | 0.98(0.54, 1.78) | 1.15(0.62, 2.14) |
| | Oromia | | | 0.42(0.23, 0.75) | 0.42(0.21, 0.81) |
| | Somali | | | 0.17(0.07, 0.42) | 0.38(0.13, 1.07) |
| | Beneshangul Gumuz | | | 0.39(0.15, 0.101) | 0.37(0.13, 1.04) |
| | SNNP | | | 0.40(0.22, 0.74) | 0.58(0.29, 1.17) |
| | Gambela | | | 0.34(0.09, 1.23) | 0.53(0.13, 2.14) |
| | Harari | | | 0.86(0.16, 4.62) | 0.65(0.11, 3.82) |
| | Addis ababa | | | 0.76(0.33, 1.74) | 0.55(0.23, 1.32) |
| | Dire dawa | | | 1.42(0.37, 5.52) | 1.03(0.24, 4.36) |
| Educational status | No education | | 1 | | 1 |
| | Primary | | 1.44(1.14, 1.81) | | 1.45(1.15, 1.84) |
| | Secondary | | 2.03(1.40, 2.96) | | 2.21(1.51, 3.24) |
| | Higher | | 2.25(1.31, 3.88) | | 2.42(1.38, 4.26) |
| Wealth index | Poorest | | 1 | | 1 |
| | Poorer | | 1.19(0.85, 1.65) | | 1.26(0.90, 1.77) |
| | Middle | | 1.45(1.03, 2.04) | | 1.51(1.06, 2.14) |
| | Richer | | 1.88(1.30, 2.70) | | 1.92(1.32, 2.80) |
| | Richest | | 3.53(2.30, 5.42) | | 3.86(2.35, 6.33) |
| Current marital status | never in union | | 1 | | 1 |
| | married | | 0.82(0.21, 3.19) | | 0.97(0.24, 3.87) |
| | living with partner | | 3.45(3.53, 22.60) | | 3.91(0.56, 27.32) |
| | widowed | | 0.90(0.0.17,4.74) | | 1.22(0.22, 6.72) |
| | divorced | | 1.95(0.46, 8.35) | | 2.31(0.53, 10.16) |
| | separated | | 2.98(0.60, 14.77) | | 3.50(0.70, 17.63) |
| Birth order | 1 | | 1 | | 1 |
| | 2–4 | | 0.96(0.72, 1.28) | | 1.03(0.77, 1.39) |
| | ≥5 | | 1.08(0.73, 1.59) | | 1.23(0.83, 1.84) |
| Number of ANC | 1 | | 1 | | 1 |
| | 2 | | 3.20(1.77, 5.80) | | 3.11(1.70, 5.68) |
| | 3 | | 5.55(3.21, 9.61) | | 5.46(3.12, 9.55) |
| | ≥4 | | 6.74(3.91, 11.63) | | 6.44(3.70, 11.21) |
| Timing of ANC visit in months | 3 | | 1 | | 1 |
| | <3 | | 0.85(0.69, 1.05) | | 0.89(0.72, 1.10 |
| | Not known | | 0.81(0.30, 2.17) | | 0.76(0.27, 2.09) |

(*Continued*)

**Table 3.** (Continued)

| Variable (n = 2918) | Category | Null Model | Model II<br>AOR (95% CI) | Model III<br>AOR (95% CI) | Model IV<br>AOR (95% CI) |
|---|---|---|---|---|---|
| ANC at government hospital | No | | | 1 | 1 |
| | Yes | | | 3.41 (2.13, 5.48) | 3.07(1.86, 5.08) |
| ANC at government health centre | No | | | 1 | 1 |
| | Yes | | | 2.39(1.67, 3.42) | 2.52(1.72, 3.68) |
| ANC at government health post | No | | | 1 | 1 |
| | Yes | | | 0.88(0.62, 1.25) | 1.08(0.75, 1.58) |
| ANC at private hospital | No | | | 1 | 1 |
| | Yes | | | 1.83(0.83, 4.04) | 1.33(0.57, 3.08) |
| ANC at NGO health facility | No | | | 1 | 1 |
| | Yes | | | 3.40(1.40, 8.26) | 4.23(1.65, 10.85) |
| Religion | Orthodox | | 1 | | 1 |
| | Catholic | | 2.29(0.32,16.36) | | 3.67(0.51, 26.29) |
| | Protestant | | 0.45(0.32,0.64) | | 0.62(0.41,0.92) |
| | Muslim | | 0.95(0.67, 1.36) | | 1.36(0.90,2.05) |
| | Traditional | | 0.21(0.05, 0.84) | | 0.35(0.09,1.41) |
| | Other | | 0.02(0.001,2.51) | | 0.02(0.001,2.94) |
| Sex of household head | Male | | 1 | | 1 |
| | Female | | 0.58(0.42,0.81) | | 0.54(0.39,0.76) |

components ranging from six to twelve. As the number of components increases their probability to receiving all gets lower. Other possible reason is that our current study used countrywide data while some others used pocket studies in different parts of the countries which in turn unable to show the real proportion of the service to country level.

On the other hand, this finding is lower than the finding of a study conducted in Myanmar (58%) [25]. The possible reason for the observed difference can be attributed to differences in socio demographic characteristics of the study participants, differences in health care system of the countries and timing of the study. Furthermore, the current study applied multilevel logistic regression which accounts for the community/higher level.

Region from where the women recruited is significantly associated with adequacy of ANC components provided. Accordingly, those from Oromia were less likely to receive adequate ANC components compared to women from Tigrai region. The finding of this study is consistent with studies conducted in Bangladesh [14] and Ethiopia [16] where administration divisions/regions showed significant association with receiving adequate component of ANC. This regional variation might be due to socio cultural differences among the regions as Ethiopia is a multicultural country. On the other hand it might be the reflection of inconsistencies in commitments of the regional health authorities to implement recommendations and guidelines and loose follow up in some of the regions.

Educational status of the participants is another variable significantly associated with receiving adequate components of ANC in which primary and above level of school attendance has positive relationship with the outcome of interest. This is in agreement with studies conducted in Bangladesh [14], East Africa [17], Zambia [21] and Ethiopia [15, 16]. This can be due to the fact that educated women have better access to information on the services given during the ANC visit and in a better position to explain their ideas and ask for what they intend to get at health facilities. Likewise more educated women are relatively empowered and can decide on their reproductive rights compared with their counterparts.

The other independent variable showed significant association with the outcome variable is household wealth index in which those with middle, richer and richest wealth index were more likely to get adequate ANC components. This is consistent with studies conducted in Bangladesh [14], Ethiopia [15, 16], East Africa [17] and Nepal [26]. The possible reason for this difference might be due to the fact that the wealthier women have the financial means to afford transportation to healthcare facilities. Additionally, women in better wealth index may be more educated and aware of the importance of receiving ANC contents, leading them to seek out and prioritize their antenatal care. In contrast, poorer women may face barriers such as lack of transportation and lack of awareness about the importance of ANC components, making it more difficult for them to access and receive the necessary ANC contents.

Number of antenatal care visit is significantly associated with receiving the components and those with more than one visit were more likely to get adequate components. This finding is consistent with the findings of studies conducted in Ethiopia [15], Harar [13], Bangladesh [14], Myanmar [25] and low and middle income countries [20]. This can be attributed to the fact that as ANC visit increases the possibility of women to be exposed to different health professionals since commitment among the health professionals might differ. In addition to this, as frequency of the visit increases the chance of receiving increases whether the health care workers provided by themselves or the women requests for the services due to her repeated visit and having chance of getting information from other clients.

On the other hand those women following their ANC at government hospital and health centres as well as at NGO health facilities were with higher odds of getting adequate component compared with their counterparts. This finding is consistent with the finding of study conducted in Bangladesh [14]. This may be due to their better access and relatively affordable cost; especially most of the services related to maternity are given for free in governmental hospitals and health centres. Furthermore, Government hospitals and health centers typically follow national guidelines and protocols for ANC, which ensures that women receive all the necessary components of care, such as regular check-ups, blood, and urine sample tests. NGO health facilities also often adhere to these guidelines.

## Strengths and limitations of the study

We used country level data that can show the level of the received components of ANC which is an indicator of the quality of the service. In addition to this, we applied multilevel statistical analysis, which shows the amount of variation contributed by the different levels of variables. On the other hand, in this study we considered only four of the ANC components since we have used secondary data from Mini Ethiopian Demographic and Health Survey and it might be considered as limitation of the study. Additionally, the study did not consider contextual factors such as the capacity of healthcare facilities, the competency and attitudes of healthcare professionals, and other sociocultural contexts that could have a greater impact on the adequacy of ANC components.

## Conclusion and recommendations

The proportion of women who adequate ANC components was slightly greater than fifty percent among women on ANC. Woman's educational status, wealth index, region of their dwelling, type of health facility, and number of ANC visits were significantly associated with receiving adequate component among the study participants. All concerned stakeholders should consider contents/components of the service instead of mere health facility visits during ANC follow up. It is important to focus on marginalized groups of women, such as those with low wealth index, when addressing this issue. Additionally, more research is needed to

explore the disparities between government and private health facilities. In addition to improving the proportion of women receiving the recommended number of ANC visit, the intended components of the care should get attention in order to meet the target of reducing maternal mortality.

## Acknowledgments

We would like to extend our appreciation to the DHS programme for the access of the dataset and special thanks for the pregnant women provided their valuable information during the demographic and Health survey.

## Author Contributions

**Conceptualization:** Desalegn Shiferaw.

**Data curation:** Desalegn Shiferaw, Bikila Regassa Feyisa, Bayise Biru, Mubarek Yesse.

**Formal analysis:** Desalegn Shiferaw, Bikila Regassa Feyisa.

**Investigation:** Desalegn Shiferaw.

**Methodology:** Desalegn Shiferaw.

**Software:** Desalegn Shiferaw, Bikila Regassa Feyisa, Bayise Biru, Mubarek Yesse.

**Writing – original draft:** Desalegn Shiferaw.

**Writing – review & editing:** Desalegn Shiferaw, Bikila Regassa Feyisa, Bayise Biru, Mubarek Yesse.

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
