## [Editor Report · Decision Letter 0]

6 Jun 2023

PONE-D-23-14411

Antenatal Care Components Utilization and Associated Factors among Pregnant women in Ethiopia: Multi level Analysis of Ethiopian Mini Demographic and Health survey 2019

PLOS ONE

Dear Dr. Shiferaw,

Thank you for submitting your manuscript to PLOS ONE. After careful consideration, we have decided that your manuscript does not meet our criteria for publication and must therefore be rejected.

I am sorry that we cannot be more positive on this occasion, but hope that you appreciate the reasons for this decision.

Kind regards,

Belayneh Mengist, MPH

Academic Editor

PLOS ONE

Additional Editor Comments:

The manuscript lacks novelty; a study conducted using the same data uncover the aim of this study (https://doi.org/10.1371/journal.pgph.0000584 ).

- - - - -

---

## [Author Response · Author response to Decision Letter 0]

1 Jul 2023

there is no single point that questions novelty of our current study compared with the mentioned article; the detail of this response is mentioned in the rebuttal letter.

---

## [Decision Letter · Decision Letter 1]

11 Jan 2024

PONE-D-23-14411R1

Antenatal Care Components Utilization and Associated Factors among Pregnant women in Ethiopia: Multi level Analysis of Ethiopian Mini Demographic and Health survey 2019

PLOS ONE

Dear Dr. Shiferaw,

Thank you for submitting your manuscript to PLOS ONE. After careful consideration, we feel that it has merit but does not fully meet PLOS ONE’s publication criteria as it currently stands. Therefore, we invite you to submit a revised version of the manuscript that addresses the points raised during the review process.

Please find the reviewers comments below and address them accordingly. 

We look forward to receiving your revised manuscript.

Kind regards,

Mohammed Feyisso Shaka, MPH

Academic Editor

PLOS ONE

2. Please upload a copy of Figures 1 and 2 which you refer to in your text on pages 19 and 20. Or if the figure is no longer to be included as part of the submission please remove all reference to it within the text.

Additional Editor Comments (if provided):

Reviewers' comments:

Reviewer's Responses to Questions

**Comments to the Author**

1. If the authors have adequately addressed your comments raised in a previous round of review and you feel that this manuscript is now acceptable for publication, you may indicate that here to bypass the “Comments to the Author” section, enter your conflict of interest statement in the “Confidential to Editor” section, and submit your "Accept" recommendation.

Reviewer #1: All comments have been addressed

Reviewer #2: All comments have been addressed

Reviewer #3: (No Response)

2. Is the manuscript technically sound, and do the data support the conclusions?

Reviewer #1: Yes

Reviewer #2: Yes

Reviewer #3: No

3. Has the statistical analysis been performed appropriately and rigorously? 

Reviewer #1: Yes

Reviewer #2: Yes

Reviewer #3: Yes

4. Have the authors made all data underlying the findings in their manuscript fully available?

Reviewer #1: Yes

Reviewer #2: Yes

Reviewer #3: No

5. Is the manuscript presented in an intelligible fashion and written in standard English?

Reviewer #1: Yes

Reviewer #2: Yes

Reviewer #3: No

6. Review Comments to the Author

Reviewer #1: • There is grammar and spelling error throughout the document. Thus, the language editing need to be carefully correct errors in grammar, punctuation, and consistency

Introduction

• The authors would do better to disclose the evidence gap in the earlier research and addressed by the current study in the introduction section.

• There are few sentences about adequate ANC in the introduction section. Better to introduce the concept of adequate ANC components and its contribution to maternal and child health in the introduction section.

Model fitness:

• The authors provided no information regarding whether or not the data was fitted to the model. Therefore, it will be beneficial to mention that the goodness of fit test taken into account and to state whether the data was fitted to a multi-level logistic regression model.

Discussion

• The novel idea in this study the adequate ANC components was not well discussed in terms of its implication for health professional, program planners, and policy makers. Others pertinent findings of the study need to be interpreted and discussed for their program and policy implication at different level.

Limitation

• The authors did not go into much detail about the limitation of the study. For instance, rather than client-related considerations, contextual factors such as the capacity of healthcare facilities and the competency of healthcare professionals have a greater impact on the adequate of ANC components. Thus, these and other study limitations must be addressed.

Reviewer #2: First, we would like to say thanks for editor who gave this chance to revise the manuscript entitled “Antenatal Care Components Utilization and Associated Factors among Pregnant women in Ethiopia: Multi level Analysis of Ethiopian Mini Demographic and Health survey 2019.” Next to this, i would like to proceed to provide few comments and suggestions for authors as follows.

On methods

1. In line 120 …Independent variables… please, classify Independent variables in to two subgroups as follow

Individual level Variables:

Community level variables:

2. In line 120 …Independent variables… if possible, you will add “sex of household head and religion” as predictor’s or Independent variables.

On results

3. In line 206 ….Factors associated with receiving ANC contents… you were employed multilevel logistic regression model. In multilevel logistic regression model, you were doing random effect and model comparison in order to select the best fitted model. Please state the results of each model in the form of the following templates. These helps readers to access the message easily and clearly.

Parameters model I model II model III Model IV

ICC (%)

Model fitness

AIC

4. What are the criteria to select the best fitted model in multilevel logistic regression model? (i.e. Model I, model II, model III and Model IV)

 

On discussion

5. Your discussion was clearly written, precise and well stated. But, in order to make your discussions strong and scientifically sounded, as much as possible try to cites some scientific literatures for your justifications or explanation throughout your discussion.

6. On line 301-303 …..”This might be because of those in better wealth status have access to better services and more confident to ask for their right”…… you were stating the above justification. Since, most ANC packages were free of charges or fees. Does the stated explanation was enough? There were several literatures that support this finding, please revise those literatures and add more justification.

Reviewer #3: Title: Antenatal Care Components Utilization and Associated Factors among Pregnant Women in Ethiopia: Multi level Analysis of Ethiopian Mini Demographic and Health survey 2019

General comments

What does Antenatal Care Components Utilization mean? And how that is different from a lot of evidence already published from the same dataset of MEDHS 2019

It looks like this study is adding another duplication of already over-duplicated information and I don’t see any reason to have it again. This is the most researched topic.

Abstract

The abstract introduction is meaningless: not mentioning why to conduct the study, no gap, no something this study going to do, and fake objective stating ‘adequate antenatal care’. I recommend rewrite

The method section should include information on multilevel model (what, why…etc)

The ANC of 55% is very from real figure from other studies which is around 48% and the pooled prevalence of 63.77%?

Positive and negative associations not separated?

The abstract conclusion do not make any sense whatsoever, redraft with right summary, practical recommendation and limitation.

Introduction

- The introduction could be clearer and more concise. Consider restructuring the section.

- Use academic writing style. Your document is full of non-academic terms e.g. ‘sluggish decline’ on paragraph 2

- It provides extensive background information about ANC without discussing and comparing the magnitude of the problem and making logical arrangements on the severity categories.

- It might be more effective if the text is more concise and focused on the most crucial information, including clearly identifying the gap in knowledge that this study is addressing. Many studies already discussed this information and it is already overwhelming. Thus, you need specific importance to continue with study.

- There is a lack of structure in presenting previous studies and findings. Organizing these findings more systematically would allow for a clearer understanding of the previous work in this area and the contribution of the current study.

- The final paragraph should more succinctly summarize the rationale for the study and the specific objectives. The current text is extensive and could be more concise and focused.

- It would be better to arrange the section with what was presented to be ANC components

- Or include enough details about each essential component you are trying to talk about.

Methods

- The method has enough details but is still highly confusing. It doesn’t contain information on ANC components. Overall, it is really difficult to understand this study and know what the authors trying to do.

- Re-write your methods based on components that we expect in the result section

- There was no detailed structure presented regarding the statistical methods

- Detailed definition of each variable not included

- Include an explanation regarding the possibility of bias and multicollinearity need to be addressed

-

Results

- The result presentation is supposed to follow the components indicated under the method section. Please, correct the method section and results accordingly.

- The interpretation of the analysis results needs to be clearer, highlighting the significance of the findings

- Provide a clearer explanation of the statistical results, ensuring that terms and statistics are understandable for readers without a statistical background.

- Ensure consistency in the presentation of results, using a uniform format for presenting statistical data and other information.

- At the end of the results section, provide a concise summary of the main findings and their significance.

Discussion

- include in-depth comparison with existing literature, especially contrasting findings and methodologies. Further, discussing how this study adds to the existing body of knowledge could be more explicit. Please, focus because this information is missing, while it is the base for understanding your study.

- Include all limitations related methods and data such as methodology used, data, sampling, representatively, inclusion criteria, or analysis methods

Conclusion

- The main conclusions of the study should be presented more clearly and concisely. It's essential to summarize the key findings and their implications straightforwardly.

- The conclusion should include more detailed recommendations for future research.

- The conclusion could elaborate more on the policy implications of the findings. How should policymakers approach

7. PLOS authors have the option to publish the peer review history of their article (what does this mean?). If published, this will include your full peer review and any attached files.

Reviewer #1: **Yes: **Simegnew Handebo (School of Public Health, St. Paul’s Hospital Millennium Medical College)

Reviewer #2: No

Reviewer #3: No

---

## [Author Response · Author response to Decision Letter 1]

23 Feb 2024

Point by point response to the reviewers’ comments

First of all, we would like to say thank you to all who have participated in reviewing our manuscript and your comments are addressed as follows.

Response: Thank you for your advice and the manuscript is prepared per PLOS ONE’s style requirement.

2. Please upload a copy of Figures 1 and 2 which you refer to in your text on pages 19 and 20. Or if the figure is no longer to be included as parts of the submission please remove all reference to it within the text.

Response: thank you, the figures are uploaded

Reviewer #1: • there is grammar and spelling error throughout the document. Thus, the language editing need to be carefully correct errors in grammar, punctuation, and consistency

Introduction

Response: Thank you for you valuable comments and we have edited the manuscript accordingly.

• The authors would do better to disclose the evidence gap in the earlier research and addressed by the current study in the introduction section.

Response: thank you for the comment and the introduction part is revised accordingly.

• There are few sentences about adequate ANC in the introduction section. Better to introduce the concept of adequate ANC components and its contribution to maternal and child health in the introduction section.

Response: Thank you for this important comment and we have added more about ANC components in the introduction.

Model fitness:

• The authors provided no information regarding whether or not the data was fitted to the model. Therefore, it will be beneficial to mention that the goodness of fit test taken into account and to state whether the data was fitted to a multi-level logistic regression model.

Response: Thank you very much for this valuable comment. We incorporated a separate table regarding model fitness in the manuscript and we hope that it addresses the issue.

Discussion

• The novel idea in this study the adequate ANC components was not well discussed in terms of its implication for health professional, program planners, and policy makers. Others pertinent findings of the study need to be interpreted and discussed for their program and policy implication at different level.

Response: Thank you for this brilliant comment and we added further elaboration and discussion regarding the ANC components adequacy.

Limitation

• The authors did not go into much detail about the limitation of the study. For instance, rather than client-related considerations, contextual factors such as the capacity of healthcare facilities and the competency of healthcare professionals have a greater impact on the adequate of ANC components. Thus, these and other study limitations must be addressed.

Response: Thank you for your comments and suggestions. We have incorporated the points raised and they are really important.

Reviewer #2: First, we would like to say thanks for editor who gave this chance to revise the manuscript entitled “Antenatal Care Components Utilization and Associated Factors among Pregnant women in Ethiopia: Multi level Analysis of Ethiopian Mini Demographic and Health survey 2019.” Next to this, i would like to proceed to provide few comments and suggestions for authors as follows.

On methods

1. In line 120 …Independent variables… please, classify Independent variables in to two subgroups as follow

Individual level Variables:

Community level variables:

Response: thank you very much for your valuable comments and suggestion. The independent variables are separately discussed per their level.

2. In line 120 …Independent variables… if possible, you will add “sex of household head and religion” as predictor’s or Independent variables.

Response: Thank you for this golden idea and comments. We rerun the model by including sex of the head of household and religion of the participants. Accordingly, we found them significant and we reported the estimates of the other independent variables considering the inclusion of the two variables.

On results

3. In line 206 ….Factors associated with receiving ANC contents… you were employed multilevel logistic regression model. In multilevel logistic regression model, you were doing random effect and model comparison in order to select the best fitted model. Please state the results of each model in the form of the following templates. These helps readers to access the message easily and clearly.

Parameters model I model II model III Model IV

ICC (%)

Model fitness

AIC

Response: we really appreciate your suggestions and comments. Per your suggestion we separately included the model diagnosis parameters as table three.

4. What are the criteria to select the best fitted model in multilevel logistic regression model? (i.e. Model I, model II, model III and Model IV)

 

Response: we selected the variable with the smallest AIC value and the final model was the best fitted model.

On discussion

5. Your discussion was clearly written, precise and well stated. But, in order to make your discussions strong and scientifically sounded, as much as possible try to cites some scientific literatures for your justifications or explanation throughout your discussion.

Response: thank you for your suggestion and comment; we have modified accordingly.

6. On line 301-303 …..”This might be because of those in better wealth status have access to better services and more confident to ask for their right”…… you were stating the above justification. Since, most ANC packages were free of charges or fees. Does the stated explanation was enough? There were several literatures that support this finding, please revise those literatures and add more justification.

Response: Thank you for your deep insight and suggestions. We have corrected it accordingly 

Reviewer #3: Title: Antenatal Care Components Utilization and Associated Factors among Pregnant Women in Ethiopia: Multi level Analysis of Ethiopian Mini Demographic and Health survey 2019

General comments

What does Antenatal Care Components Utilization mean? And how that is different from a lot of evidence already published from the same dataset of MEDHS 2019

It looks like this study is adding another duplication of already over-duplicated information and I don’t see any reason to have it again. This is the most researched topic.

Response: Thank you for your concern and comment. We have reviewed if there is a similar study from the same dataset and found no similar study with objective. Rather, other articles addressed the area diffrently

Abstract

The abstract introduction is meaningless: not mentioning why to conduct the study, no gap, no something this study going to do, and fake objective stating ‘adequate antenatal care’. I recommend rewrite

Response: thank you for your comment and we rewrote the introduction part of the article by incorporating additional points

The method section should include information on multilevel model (what, why…etc)

The ANC of 55% is very from real figure from other studies which is around 48% and the pooled prevalence of 63.77%?

Response: Thank you for your comment. The reason it is 55% in this study is that we have considered only four of the components namely blood pressure measure, blood and urine sample taking and iron supplementation.

Positive and negative associations not separated?

Response: Thank you for your comment and the point is addressed accordingly.

The abstract conclusion do not make any sense whatsoever, redraft with right summary, practical recommendation and limitation.

Response: thank you for your suggestion and it is addressed accordingly.

Introduction

- The introduction could be clearer and more concise. Consider restructuring the section.

- Use academic writing style. Your document is full of non-academic terms e.g. ‘sluggish decline’ on paragraph 2

Response: thank you for the comment and it is addressed 

- It provides extensive background information about ANC without discussing and comparing the magnitude of the problem and making logical arrangements on the severity categories.

- It might be more effective if the text is more concise and focused on the most crucial information, including clearly identifying the gap in knowledge that this study is addressing. Many studies already discussed this information and it is already overwhelming. Thus, you need specific importance to continue with study.

Response: Thank you for the comments and suggestions. The points are addressed accordingly.

- There is a lack of structure in presenting previous studies and findings. Organizing these findings more systematically would allow for a clearer understanding of the previous work in this area and the contribution of the current study.

Response: thank you for the comment and we have addressed them

- The final paragraph should more succinctly summarize the rationale for the study and the specific objectives. The current text is extensive and could be more concise and focused.

- It would be better to arrange the section with what was presented to be ANC components

- Or include enough details about each essential component you are trying to talk about.

Response: thank you for your valuable comments and insights and the points are addressed.

Methods

- The method has enough details but is still highly confusing. It doesn’t contain information on ANC components. Overall, it is really difficult to understand this study and know what the authors trying to do.

Response: thank you and we addressed the comments

- Re-write your methods based on components that we expect in the result section

Response: thank you and the comments are incorporated.

- There was no detailed structure presented regarding the statistical methods

- Detailed definition of each variable not included

- Include an explanation regarding the possibility of bias and multicollinearity need to be addressed

Response: thank you for your insights and comments. We have addressed your comments

-

Results

- The result presentation is supposed to follow the components indicated under the method section. Please, correct the method section and results accordingly.

Response: Thank you for the suggestion and we have modified it accordingly.

- The interpretation of the analysis results needs to be clearer, highlighting the significance of the findings

Response: thank you for your comment and we have addressed it

- Provide a clearer explanation of the statistical results, ensuring that terms and statistics are understandable for readers without a statistical background.

Response: thank you for your valuable comments and we have modified the specific area accordingly 

- Ensure consistency in the presentation of results, using a uniform format for presenting statistical data and other information.

Response: thank you and we have improved it accordingly

- At the end of the results section, provide a concise summary of the main findings and their significance.

Response: thank you for your time and valuable comments. We have included it accordingly

Discussion

- include in-depth comparison with existing literature, especially contrasting findings and methodologies. Further, discussing how this study adds to the existing body of knowledge could be more explicit. Please, focus because this information is missing, while it is the base for understanding your study.

Response: Thank you for the comments and we have addressed it

- Include all limitations related methods and data such as methodology used, data, sampling, representatively, inclusion criteria, or analysis methods

Response: thank you for your valuable comments and it is well addressed 

Conclusion

- The main conclusions of the study should be presented more clearly and concisely. It's essential to summarize the key findings and their implications straightforwardly.

- The conclusion should include more detailed recommendations for future research.

- The conclusion could elaborate more on the policy implications of the findings. How should policymakers approach

Response: Thank you for your golden comments and we have incorporated them accordingly.

---

## [Decision Letter · Decision Letter 2]

26 Mar 2024

PONE-D-23-14411R2Antenatal Care Components Utilization and Associated Factors among Pregnant women in Ethiopia: Multi level Analysis of Ethiopian Mini Demographic and Health survey 2019PLOS ONE

Dear Dr. Shiferaw,

Thank you for submitting your manuscript to PLOS ONE. After careful consideration, we feel that it has merit but does not fully meet PLOS ONE’s publication criteria as it currently stands. Therefore, we invite you to submit a revised version of the manuscript that addresses the points raised during the review process.

Your manuscript still needs minor but essential revisions. The language needs extensive revision. There is inconsistency in font size and style.Full of barely understandable expressionsSome of the comments of the reviewers were also not addressed adequately.Further essential comments using the track change on the attached file by the editor. ==============================

We look forward to receiving your revised manuscript.

Kind regards,

Mohammed Feyisso Shaka, MPH

Academic Editor

PLOS ONE

Journal Requirements:

Reviewers' comments:

Reviewer's Responses to Questions

**Comments to the Author**

1. If the authors have adequately addressed your comments raised in a previous round of review and you feel that this manuscript is now acceptable for publication, you may indicate that here to bypass the “Comments to the Author” section, enter your conflict of interest statement in the “Confidential to Editor” section, and submit your "Accept" recommendation.

Reviewer #3: All comments have been addressed

2. Is the manuscript technically sound, and do the data support the conclusions?

Reviewer #3: Yes

3. Has the statistical analysis been performed appropriately and rigorously? 

Reviewer #3: Yes

4. Have the authors made all data underlying the findings in their manuscript fully available?

Reviewer #3: Yes

5. Is the manuscript presented in an intelligible fashion and written in standard English?

Reviewer #3: Yes

6. Review Comments to the Author

Reviewer #3: The authors addressed all my concerns, I do have no more comments. I am greatly appreciative of the responses.

7. PLOS authors have the option to publish the peer review history of their article (what does this mean?). If published, this will include your full peer review and any attached files.

Reviewer #3: **Yes: **Girma Gilano

---

## [Author Response · Author response to Decision Letter 2]

2 Apr 2024

Point by point response to the academic editor’s comments

First of all, I would like to say thank you for your essential comments and your comments are addressed as follows.

1. Comments 1 and 2: What is this number?

Response: thank you for your critical observation and valuable comments. These numbers where the level of association I calculated during the previous version. Later on after the reviewers comment, I included religion and sex of the head of the household in multivariable analysis after which the value of AOR was changed. These numbers where remained from the former version by my own error and now I have corrected it. Thank you

2. Comment 3: Your write up here is not understandable. Please revise

Response: Thank you for your constructive comment and now I have revised it.

3. Comment 4: Yes that is well known. So how it should be improved? Any possible recommendation from the implication of your study?

Response: thank you for your comment and I have included possible recommendations based on the findings

4. comment 5: Write in standard form

Response: thank you for your comment and it is corrected now.

Comment 6: What is the difference between this statement and the previous one?

Response: Thank you for your meticulous comment and corrected by removing the duplicated statement.

5. Comment 7: For which visit was the outcome measured? Not clear

Response: Thank you for your comment. The outcome of interest including the four components was considered irrespective of the order or number of the visit. It can be at any of the visits.

6. Comment 8: Please group the figure to connect the boxes together and submit it as a supplementary file.

Response: Thank you for your comment and I have submitted as supplementary file per your comment.

7. Comment 9: But you operationalized as Yes/No, see above. When you say it is adequate?

Response: when it is adequate it is “Yes” and when it is in adequate it is “ No”

8. Comment 10: Please use “≥” or “≤”for accordingly

Response: thank you for your comment and it is corrected accordingly. 

9. Comment 11: The way you interpret the results is confusing and not understandable. Please revise it

Response: Thank you for your comment and it is revised now.

10. Comment 12: Any potential competing interests, Funding information, Author contributions

Response: Thank you for your comment and these components are included in the manuscript 

11. Comment 13: Please review your reference list to ensure that it is complete and correct

Response: thank you for your comment. I have excluded the 11th reference on the former version since it is not this much relevant and the remaining references are enough to support the specific finding.

---

## [Editor Report · Decision Letter 3]

18 Apr 2024

Antenatal Care Components Utilization and Associated Factors among Pregnant women in Ethiopia: Multi level Analysis of Ethiopian Mini Demographic and Health survey 2019

PONE-D-23-14411R3

Dear Dr. Shiferaw,

We’re pleased to inform you that your manuscript has been judged scientifically suitable for publication and will be formally accepted for publication once it meets all outstanding technical requirements.

Kind regards,

Mohammed Feyisso Shaka, MPH

Academic Editor

PLOS ONE
---

## [Editor Report · Acceptance letter]

17 May 2024

PONE-D-23-14411R3 

PLOS ONE

Dear Dr. Shiferaw, 

I'm pleased to inform you that your manuscript has been deemed suitable for publication in PLOS ONE. Congratulations! Your manuscript is now being handed over to our production team.

Kind regards, 

on behalf of

Mr. Mohammed Feyisso Shaka 

Academic Editor

PLOS ONE